# Structural effects of the highly protective V127 polymorphism on human prion protein

Laszlo L. P. Hosszu [1], Rebecca Conners[1,2,6], Daljit Sangar[1], Mark Batchelor [1], Elizabeth B. Sawyer [1,7], Stuart Fisher[3,8], Matthew J. Cliff [4], Andrea M. Hounslow[5], Katherine McAuley[3], R. Leo Brady [2], Graham S. Jackson[1], Jan Bieschke[1], Jonathan P. Waltho [4,5] & John Collinge [1]✉

Prion diseases, a group of incurable, lethal neurodegenerative disorders of mammals including humans, are caused by prions, assemblies of misfolded host prion protein (PrP). A single point mutation (G127V) in human PrP prevents prion disease, however the structural basis for its protective effect remains unknown. Here we show that the mutation alters and constrains the PrP backbone conformation preceding the PrP β-sheet, stabilising PrP dimer interactions by increasing intermolecular hydrogen bonding. It also markedly changes the solution dynamics of the β2-α2 loop, a region of PrP structure implicated in prion transmission and cross-species susceptibility. Both of these structural changes may affect access to protein conformers susceptible to prion formation and explain its profound effect on prion disease.

[1] MRC Prion Unit at UCL, UCL Institute of Prion Diseases, 33 Cleveland Street, London W1W 7FF, UK. [2] University of Bristol, School of Biochemistry, Biomedical Sciences Building, University Walk, Clifton BS8 1TD, UK. [3] Diamond Light Source, Diamond House, Harwell Science and Innovation Campus, Didcot, Oxfordshire OX11 0DE, UK. [4] Manchester Institute of Biotechnology, University of Manchester, 131 Princess Street, Manchester M1 7DN, UK. [5] Department of Molecular Biology and Biotechnology, University of Sheffield, Firth Court, Western Bank Sheffield S10 2TN, UK. [6] Present address: Living Systems Institute, University of Exeter, Stocker Road, Exeter EX4 4QD, UK. [7] Present address: London School of Hygiene & Tropical Medicine, Keppel Street, London WC1E 7HT, UK. [8] Present address: ESRF, 71, Avenue des Martyrs, CS 40220, 38043 Grenoble Cedex 9, France. ✉email: jc@prion.ucl.ac.uk

Prion diseases, such as bovine spongiform encephalopathy (BSE) in cattle, scrapie in sheep, and Creutzfeldt-Jakob disease (CJD), kuru and Gerstmann-Sträussler-Scheinker (GSS) syndrome in humans, are a group of neurodegenerative disorders caused by prions, self-replicating β-sheet-rich infectious polymeric assemblies of misfolded host-encoded cellular prion protein (PrP[C])[1–4]. Whilst rare, prion diseases are an area of intense research interest, as it is increasingly recognised that other degenerative brain diseases, such as Alzheimer's and Parkinson's diseases, also involve the accumulation and spread of aggregates of misfolded host proteins through an analogous process of seeded protein polymerisation[2,5–8]. Consequently, study of 'prion-like' mechanisms has been recognised to have much a wider relevance to the understanding of neurodegenerative disorders[9–11].

PrP[C] is a cell surface, predominantly α-helical, glycosylphosphatidylinositol (GPI)-anchored glycoprotein that is sensitive to protease treatment and soluble in detergents[1]. In contrast, prions may acquire protease-resistance and are classically designated as PrP[Sc] (refs. [12,13]). PrP[Sc] is found only in prion-infected tissue and is β-sheet-rich aggregated material, partially resistant to protease treatment, and insoluble in detergents[14].

Transmission experiments to transgenic mice provide strong supporting evidence that alternative conformers or assembly states of PrP[Sc] encode multiple prion strains, which differ in their pathogenic properties[2,15]. Transgenic mice expressing only human PrP with either valine or methionine at residue 129 have shown that this common human polymorphism constrains the propagation of distinct human prion conformers, and the occurrence of associated patterns of neuropathology consistent with the conformational selection model of prion propagation[16–20]. Heterozygousity at codon 129 is thought to confer resistance to prion disease by inhibiting homologous protein–protein interactions essential for efficient prion replication with the presence of methionine or valine at residue 129 controlling the propagation of distinct human prion strains[2,21]. Biophysical measurements suggest that this powerful effect of residue 129 on prion strain selection is likely to be mediated via its effect on the conformation of the disease-associated PrP[Sc] form, or its precursors or on the kinetics of their formation, as it has no measurable effect on the structure, folding or stability of PrP[C] [22].

The acquired prion disease kuru, which was epidemic amongst the Fore linguistic group of the Papua New Guinea highlands when first studied in the 1950′s, and which was transmitted during mortuary feasts, imposed strong genetic selection on the Fore, essentially eliminating residue 129 homozygotes[23]. A novel variant of prion protein, V127, unique to the affected population in the epicentre of the kuru epidemic, was also identified[24]. In this variant, the glycine at residue 127, which is fully conserved amongst vertebrate PrP primary structures, is substituted by valine. The V127 polymorphism was found on one copy of the *PRNP* gene in unaffected individuals within the population, suggesting that this polymorphism conferred resistance to prion disease, having been selected for in response to the kuru epidemic[23,24]. The protection afforded by this polymorphism was modelled using transgenic mice expressing human PrP[25], and showed that heterozygous mice expressing both alleles containing glycine and valine at residue 127 (G/V127), echoing the human resistance genotype, exhibited profoundly reduced susceptibility to infection with kuru and classical CJD prions. Most importantly, however, and in complete contrast to the protective effect of the residue 129 polymorphism, homozygous mice expressing human PrP with solely valine at residue 127 (V127), showed total resistance to all inoculated human prion strains. A comparison of the incubation periods between hemizygous mice expressing wild-type G127 human PrP only, with heterozygous mice expressing both G127 and V127 PrP, indicated a dose-dependent dominant-negative inhibitory effect of V127 PrP on prion propagation, resulting in prolonged incubation periods and variable attack rates in heterozygotes[25]. These data indicated that V127 PrP is intrinsically resistant to prion propagation and can inhibit propagation involving wild-type (WT) G127 PrP. In essence, this single amino acid substitution, at a residue completely conserved in vertebrate evolution, has as potent a protective effect on the host as a null mutation. Consequently, the structural and mechanistic basis of the protective effect of the V127 mutation is of keen interest as it may provide key insights into the mechanism of prion conformational conversion and recruitment.

As a first step in characterising the effect of this protective polymorphism on PrP, we undertook a detailed investigation of the effect of the residue 127 polymorphism on the biophysical properties of the native cellular PrP[C] conformation using a combination of X-ray crystallography, NMR and equilibrium unfolding. We show that this mutation imposes local changes in backbone conformation which facilitate formation of intermolecular hydrogen bonds between native-state dimers and imposes conformational restrictions on this region of the protein. In addition, it significantly alters millisecond timescale conformational rearrangements in regions of PrP proposed to be important in prion transmission[26–28]. These effects may modulate the conversion of native PrP[C] to a disease-associated form or on pathway intermediates relevant to the disease process, and provide a mechanistic explanation for the protective effect of this mutant.

## Results

**Choice of PrP variants studied**. Persons who were exposed to kuru and survived the epidemic were predominantly heterozygotes at PrP residue 129[23]. The V127 protective polymorphism in human PrP was always present on an M129 allele[24], consequently our main interest was with the V127/M129 PrP variant. However, we took the opportunity, given the known biological effect of the residue 129 polymorphism to also study the V127 variant with valine at residue 129 (V127/V129), and both forms of wild-type PrP (G127/M129 and G127/V129) with the aim of dissecting the effects of both of these protective polymorphisms.

**V127 PrP structures closely resemble wild-type G127 PrP**. To determine whether the overall structure of PrP[C] was affected by the protective V127 variant we crystallised recombinant human PrP (residues 119–231), with valine at residue 127, (V127/M129 and V127/V129), complexed with the Fab fragment of the anti-PrP antibody ICSM18, as performed previously with G127/M129 PrP (Supplementary Table 1 and Supplementary Fig. 1)[29]. The crystal structures of both V127 variants (V127/M129, 2.3 Å resolution, pdb 6SV2 and V127/V129, 2.5 Å resolution, pdb 6SUZ) closely resembled that of WT G127/M129 (pdb 2W9E, Fig. 1a and Supplementary Fig. 2)[29]. The structured C-terminal domain (residues 125–225) comprises three α-helices (α1–α3) and a short, two-stranded, anti-parallel β-sheet (Fig. 1 and Supplementary Fig. 3). Residue 127 immediately precedes the first β-strand of the β-sheet whereas residue 129 lies within it. The residues surrounding 127 and 129 are well defined in both crystal structures (Figs. 2 and 3) and show that the side-chains of both residues are predominantly located on the protein surface. Neither the 127 nor 129 polymorphisms substantially perturb the backbone or sidechain positions, or hydrogen bonding, of residues within the β-sheet (Fig. 1b and Supplementary Fig. 2a–c). Both circular dichroism (CD) and heteronuclear NMR spectra (Supplementary Figs. 4–6) are consistent with the crystal structures accurately reflecting the solution structure of the proteins. The global stability and unfolding behaviours of the V127/M129

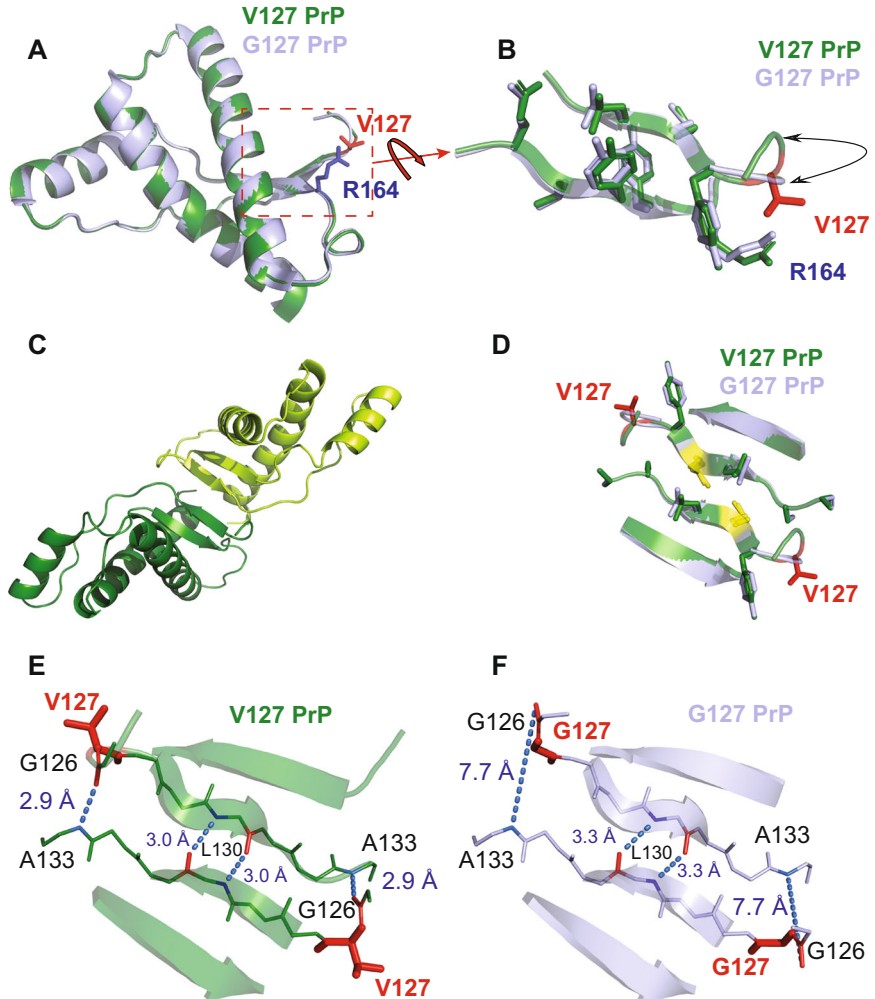

**Fig. 1 Effect of the V127 polymorphism on the structure of human PrP$^C$. a** V127/M129 (PDB 6SV2 – green) and wild-type G127/M129 human PrP (PDB 2W9E – light blue[29]) crystal structures, superimposed in cartoon representation. Residues 125–223 are shown. The r.m.s. deviations of backbone heavy atoms are less than 0.44 Å between these structures. The sidechains of V127 (red) and R164 (blue) are shown as sticks. This figure and the other structural figures were prepared using *PyMOL* (PyMOL Molecular Graphics System, Schrödinger, LLC). **b** Side chain packing in the V127/M129 (green) and WT G127/M129 (light blue) β-sheets. The PrP backbone immediately preceding residue 127 in V127/M129 PrP is displaced due to the bulkier valine sidechain at residue 127. The sidechain and backbone positions of residues in the β-sheet are very similar, with the exception of the sidechain of arginine 164 (R164), which due to its close proximity to residue 127 is displaced in the V127 variant. This perturbation (see also Fig. 8) is observed in solution by a marked chemical shift change in the Nε peak arising from the R164 sidechain group in NMR HSQC spectra (Supplementary Fig. 6). **c** Four-stranded intermolecular anti-parallel β-sheet formed between neighbouring V127/M129 PrP molecules (in green and lime green). **d** Intermolecular β-sheet contacts in V127/M129 PrP (green) and WT G127/M129 PrP (light blue). The amino acid sidechains of residues found in the intermolecular β-sheet are shown in stick representation, with the residue 127 and 129 polymorphisms in red and yellow respectively. **e, f** Intermolecular β-sheet hydrogen bonding in V127/ M129 (**e**) and G127/M129 PrP (**f**). Hydrogen bonds stabilising the intermolecular β-sheet are shown as blue dotted lines, between the amide (blue) and carbonyl (red) groups of the denoted amino acids, with the corresponding distances in Å. The β-sheet interface in the V127/M129 PrP crystal is stabilised by an additional pair of hydrogen bonds between the carbonyls of G126 and amides of A133 (**e**). The additional hydrogen bond pair between G126 and A133 is not formed in WT G127/M129 PrP as the hydrogen bond distance is too long (7.7 Å) (**f**).

and V127/V129 variants (Supplementary Fig. 7 and Supplementary Table 2) are also not significantly affected by the substitution of valine for glycine at position 127, reflecting the lack of major structural perturbation.

**V127 polymorphism restricts PrP backbone conformation.** Despite the crystal structures being mostly unperturbed by the V127 polymorphism, a number of localised differences were identifiable. The most significant area of variation is found immediately N-terminal of residue 127 (residues 125–127). This region adopts an essentially identical conformation in both the V127/M129 and V127/V129 structures, which differs significantly

from WT G127/M129 PrP (Fig. 1b and Supplementary Fig. 2a–c). In particular, the reduction in the conformational plasticity of the backbone due to the valine/glycine substitution at position 127 leads to a very different conformation at this point (V127 Phi angle = −70.5°, c.f. G127 = +106.9°), as the WT backbone conformation is in a disallowed region of conformational space for valine. Consequently, the Cα of G126 in V127/M129 PrP is displaced by 2.9 Å, and the Cα of L125 by 2.2 Å (equivalent Cα atoms of most other surrounding residues are displaced by 0.2–0.3 Å). These Cα atom positions are well defined in both V127 structures (Figs. 2 and 3).

Furthermore, the V127 polymorphism appears to reduce conformational variability at residue 127, and concomitantly

increases structural definition of the β-sheet, as implied by a comparison of relative B-factors of this region in V127/M129 PrP when compared with WT G127/M129 PrP. These lower B-factors extend from L125 to A133, beyond the end of the first strand of the β-sheet (Fig. 3). In V127/M129 PrP, the average Cα B-factors for both the N-terminus (residues 126–131; 30 Å$^2$), and β-strand 1 (residues 128–131; 27 Å$^2$) are lower than the average B-factor for the core secondary structure elements (31 Å$^2$). In contrast, in wild-type G127/M129 PrP, the corresponding values for both the N-terminus (46 Å$^2$) and β-strand 1 (42 Å$^2$) are higher than the average B-factor (39 Å$^2$). As the crystal structures are all isomorphous with the same crystal packing, we suggest that the reduction in B-factors is likely due to conformational restriction introduced by the valine sidechain, and by additional inter-molecular hydrogen bonding found in the V127 crystals, described below.

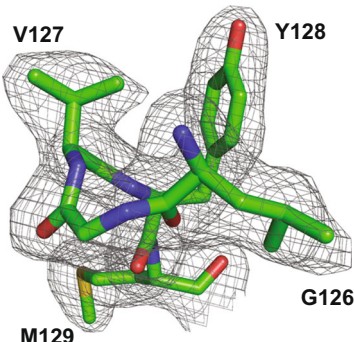

**Fig. 2 The quality of the electron density maps for PrP in the V127/M129 PrP - ICSM-18 Fab complex at 2.3 Å resolution.** Residues from the PrP β-sheet and the V127 polymorphism are shown; 2F$_O$ –F$_C$ map contoured at 1σ.

**V127 polymorphism extends PrP intermolecular β-sheet.** Notably, dimers between crystallographically-related PrP molecules are observed in the crystals (Fig. 1c). Association is mediated by a short segment of the anti-parallel β-sheet with hydrogen bonds formed between the first β-strand (residues 128–131) of each molecule[29]. This results in the formation of a four-stranded intermolecular β-sheet between the existing anti-parallel β-sheets of each PrP molecule, involving close homotypic contacts at L130 (Fig. 1e, f). Similar intermolecular interactions are also observed in the non-isomorphous crystal structures of sheep[30], rabbit[31] and human PrP[32] in the absence of antibody, and in different crystallographic space groups (Fig. 4). This suggests that this interaction is not a crystal packing artefact, and may reflect a greater biological significance for prion propagation, especially as residue 129 is protective and crucial to the aetiology and neuro-pathology of prion disease, and residue 127, which is in close proximity to the dimer interface, can completely prevent prion propagation.

The residue 129 polymorphism is accommodated within the dimer interface without significant perturbations of surrounding amino acids (Supplementary Fig. 2c, d). In contrast, substitution by valine at residue 127 results in the formation of an additional pair of intermolecular hydrogen bonds in both V127 structures, between the backbone carbonyl and amide groups of G126 and A133 respectively (Fig. 1e, Supplementary Fig. 2d), due to the alteration in backbone conformation. This orients the G126 carbonyl group towards the dimer interface, and its hydrogen bond acceptor A133. In the WT G127/M129 PrP dimer, the corresponding G126 CO – A133 N$^H$ distance is 7.7 Å, as the carbonyl group of G126 points away from the dimer interface (Fig. 1f). This additional hydrogen bonding with V127 extends the β-sheet dimer interface to residues 126–133, thereby encompassing V127, whereas G127 is not involved in dimer contacts in WT G127/M129 PrP. The hydrogen bond distances for these additional H-bonds in the V127 structures (2.8–2.9 Å) indicate a strong interaction. Also, the hydrogen bonds involving

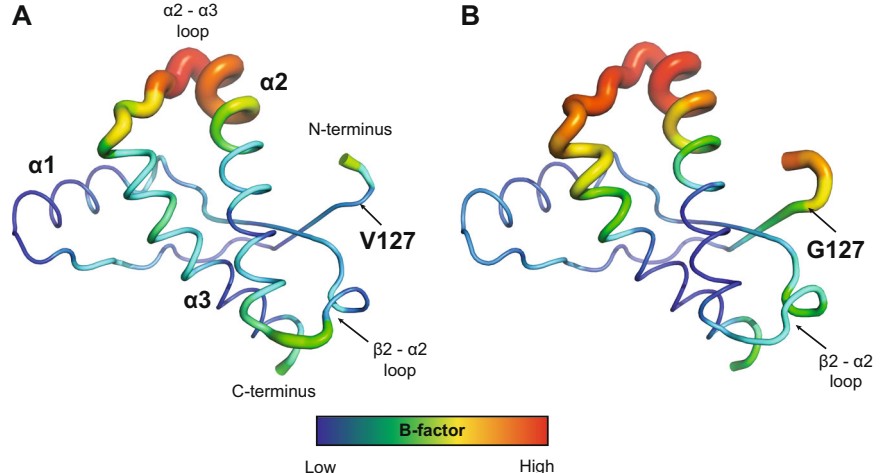

**Fig. 3 Thermal parameter (B-factor) distribution in human PrP. (a)** V127/M129 PrP **(b)** G127/M129 PrP shown as "putty" representation, as implemented by PyMOL. The V127/M129 PrP Cα atom B-factors range from 22.7 Å$^2$ to 96.9 Å$^2$ with average values of 38.3 Å$^2$ for the whole protein, and 30.4 Å$^2$ for the core secondary structure elements (residues 128–131 (β-strand 1), 144–154 (α-helix 1), 160–164 (β-strand 2), 174–186 (α-helix 2) and 202–220 (α-helix 3)). The Cα B-factors are depicted on the structure in dark blue (lowest B-factor) through to red (highest B-factor), with the radius of the ribbon increasing from low to high B-factor. The lowest B-factor is observed in the region of α-helix 2 (α2) and α-helix 3 (α3) where the disulphide bridge links the two α-helices at residues 179 and 214 (dark blue), with the antibody-binding epitope spanning α-helix 1 also displaying lower than average B-factors, consistent with the antibody contacts stabilising this region of PrP relative to the overall structure. The largest B-factors are observed in the loop region linking helices α2 and α3 (red) (α2- α3 loop; residues 191–199), where the electron density clearly shows more disorder than elsewhere in the structure. In contrast, the B-factors for residues in close proximity to the V127 polymorphism are not unusually high, and all of these residues are clearly observed in the electron density (see also Fig. 2).

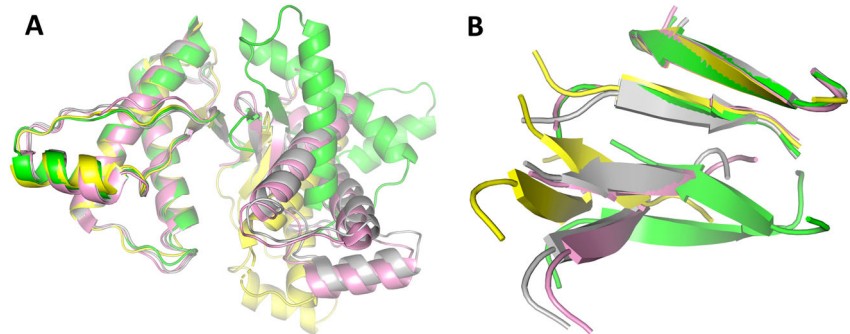

**Fig. 4 The interaction of PrP molecules in various PrP crystal structures. a** Superposition of human V127 (green), ovine[30] (pink), rabbit[31] (grey) and human D178N[32] (yellow) PrP dimers from their respective crystal structures. Unlike V127, the other structures were obtained from apo-crystals in the absence of antibody. The ICSM18 antibody-binding epitope consists of α-helix 1 which is remote from the PrP dimer interface (see **b** and Supplementary Fig. 1). **b** Close up view of the β-sheet dimer interface common to the crystal dimers. The relative orientation of the two interacting PrP molecules in each structure differs depending on the intermolecular hydrogen-bonding patterns.

L130 at the centre of the intermolecular β-sheet interface are shortened (3.0 Å compared with 3.3 Å) (Fig. 1e, f). Thus, rather than preventing PrP dimerisation through disruption of intermolecular hydrogen bonding[33,34], the V127 polymorphism appears to increase native-state dimer hydrogen bonding.

**Increased conformational variability in V127 PrP structures.** Intriguingly, altered conformational variability is observed in key regions distant from the site of the V127 polymorphism, in particular the loop linking the second strand of the β-sheet and helix 2 (β2-α2 loop; residues 165–172; Fig. 3). This region, which has been shown to affect prion cross-species transmissibility[26–28,35,36], is adjacent to the β-sheet, packing against residues N-terminal to the first β-strand (including residue 127), the β-sheet itself, the C-terminus of helix 3, and is in close proximity to the disease-associated residue D178[37]. The Cα B-factors for residues 169–172 within this loop are higher than the average for the rest of the protein in both V127 structures (50 vs. 38 Å$^2$ for V127/M129 & 56 vs. 45 Å$^2$ for V127/V129). This contrasts with WT G127/M129, where these residues are better defined than the average, according to their B-factors (40 vs. 42 Å$^2$). The B-factors for the V127 structures are consistent with an alteration in the degree of conformational exchange in this loop region of the protein compared to WT PrP, possibly compensating for the reduced conformational variability observed in the β-sheet region. The remaining regions of the V127 PrPs display B-factors that are comparable to WT values.

**Altered conformational variability also seen in solution.** Crucially, the altered conformational variabilities are also observed in solution. The effects of the V127 and V129 polymorphisms on the dynamics of PrP were investigated using NMR relaxation data (Supplementary Fig. 8), coupled with *Modelfree* (Fig. 5) and reduced spectral density analyses (Fig. 6, Supplementary Fig. 9)[38,39]. The former uses order parameters ($S^2$) to report internal sub-nanosecond (ns) motions. $S^2$ values range from 0 for highly flexible to 1 for rigid systems. The β-sheet and helical regions of all PrP variants exhibit $S^2$ values of 0.8–0.9, typical of structured regions of folded proteins (Fig. 5). However, a number of residues in structured regions, for example E168 in V127/M129 and D178 in G127/M129 PrP display anomalously low $S^2$ values. These are subject to millisecond (ms) conformational dynamics described below (Fig. 6).

The *Modelfree* approach also allows a general separation for each residue of ms conformational dynamics ($R_{ex}$ values) from ns and sub-ns motions. These ms timescale motions are often associated with large-scale co-operative conformational changes

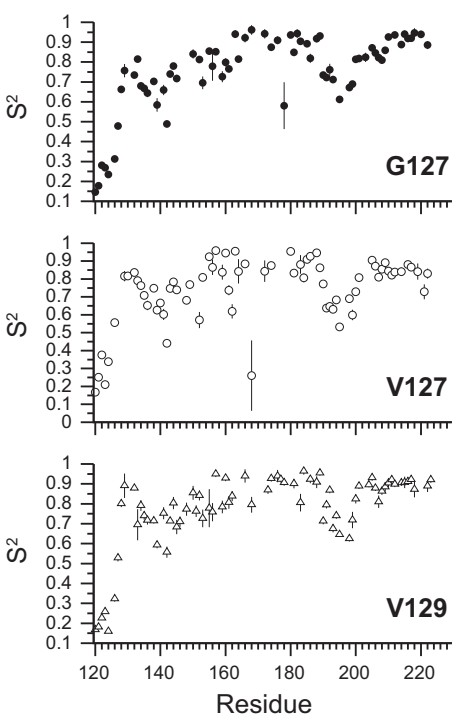

**Fig. 5 Degree of order in PrP variants.** Order parameters ($S^2$) for G127 (G127/M129), V127 (V127/M129) and V129 (G127/V129) PrP. Residues of the β1-α1 (residues 134–144) and α2-α3 loops (residues 194–199) display slightly reduced $S^2$ values (0.6–0.8), reflecting increased flexibility, commonly observed in loop regions of globular proteins[22,71,73], and in previous studies of PrP$^C$. The N-terminus (residues 119–124) preceding the β-sheet is mobile and disordered, with low $S^2$ values (0.15–0.5) and a lack of electron density in the crystal structures. These order parameters are mapped onto the structures of the PrP variants in Supplementary Fig. 10.

and highlight residues that populate low-free energy alternative conformations. For each of the PrP variants, a number of residues exhibited $R_{ex}$ values (Fig. 6c). These are concentrated in a spatially close region, involving the β-sheet (V129/G131/R164), the β2-α2 loop (M166/E168/Q172) and the C-terminus of helix 3 (I215 T216/Y218/E219/E221; Supplementary Figs. 3, 10 and 11). The line-broadening of resonances D167, Y169, S170 and N171 beyond detection in the HSQC spectra of all three variants also likely reflect ms dynamics. The observed conformational

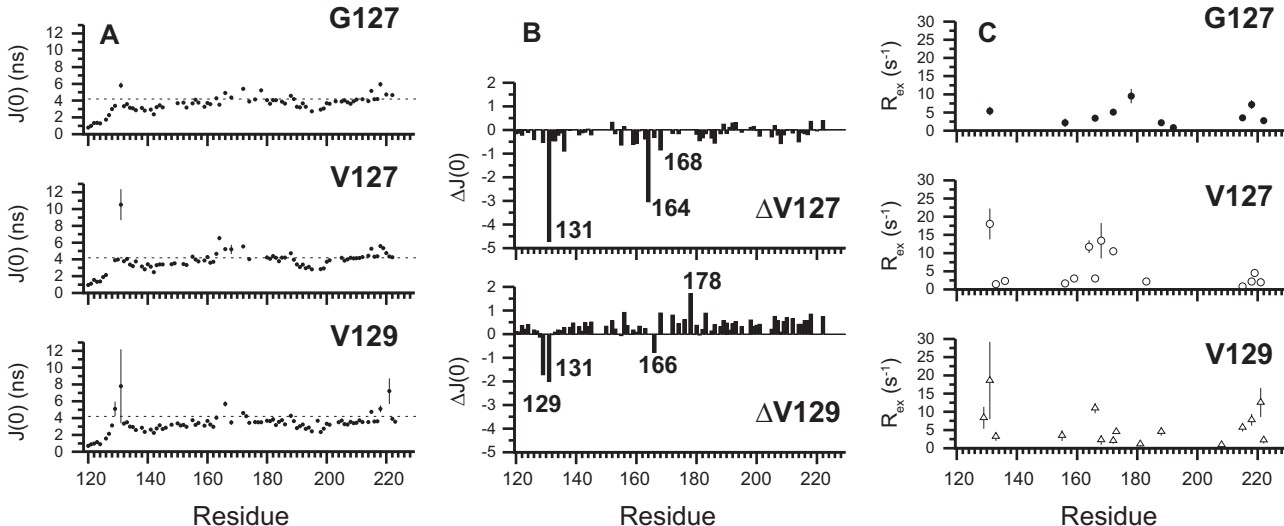

**Fig. 6 Conformational dynamics in the PrP variants. a** Reduced spectral density function J(0), describing the amplitude of zero frequency motions in the PrP variants at 800 MHz. Uncharacteristically large J(0) values, such as those exhibited by β-sheet residues (128–131 and 160–164), and G131 and R164 in V127/M129 PrP in particular, indicate ms – μs dynamics. The dotted lines in the J(0) graphs are two standard deviations greater than the mean J(0) for the N-terminus (residues 200–210) of helix 3, of the respective variants. **b** Effect of V127 and V129 polymorphisms on the amplitude of zero frequency J(0) motions at 800 MHz. J(0) changes relative to G127/M129 PrP. Residues which experience significant changes in J(0) due to the V127 substitution include G131, R164 and E168. V129 results in altered J(0) motions for residues 129 and 131, within the first β-strand, M166, and residue 178, located in helix 2. The observed changes are due to differential ms conformational dynamics (see **c**). **c** PrP ms dynamics ($R_{ex}$) modelled in the Relax Modelfree analysis. The V127 polymorphism increases ms dynamics ($R_{ex}$) within the β-sheet (G131/R164) and β2-α2 loop (E168/Q172), and diminishes those at the C-terminus of helix 3. The V129 polymorphism also increases ms dynamics in the first β-strand (V129/G131), and the C-terminus of helix 3. Residues 166 and 172 at either end of the β2-α2 loop are also perturbed. These $R_{ex}$ values are mapped onto the structure of PrP$^C$ (See Fig. 7. and also Supplementary Figs. 10 and 11).

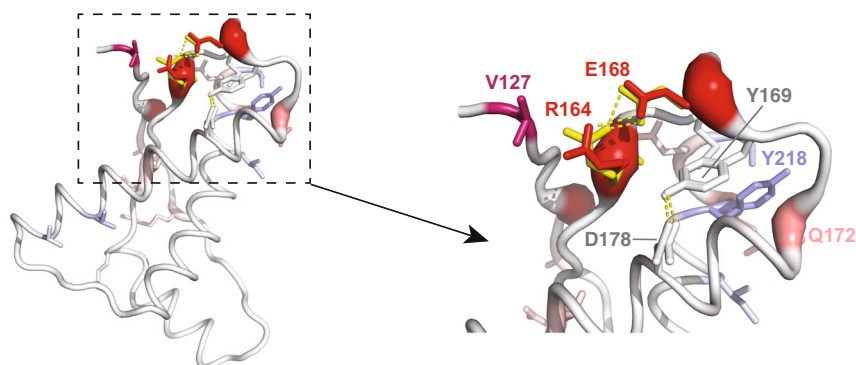

**Fig. 7 Effect of V127 polymorphism on the amplitude of PrP ms dynamics ($R_{ex}$).** The sidechains of residues which experience altered ms dynamics in V127/M129 PrP, relative to G127/M129 PrP are shown (Fig. 6c), with varying width of backbone and colour. Residues showing increased $R_{ex}$ values in the V127 variant, such as G131, R164, E168 and Q172, are coloured red, with those showing a reduction, such as Y218, are coloured blue. Those residues for which a comparison is not possible, due to absence of data are not coloured. The orientation of R164 and E168 sidechains in G127/M129 PrP are shown in yellow[29], illustrating the loss of hydrogen bonding in V127/M129 PrP, caused by a steric clash between V127 and R164 sidechains in the V127 variant. Also shown is the hydrogen bond between Y169 and D178, showing the close association between the β2-α2 loop and another residue which has a key effect on the aetiology of human prion disease.

dynamics are consistent with a proposed interconversion of the β2-α2 loop between a more populated $3_{10}$-helix and a type I β-turn[27,30,32,40,41] (Supplementary Fig. 12). The V127 polymorphism results in large increases in ms dynamics for residues G131 and R164 in the β-sheet, and E168 and Q172 in the β2-α2 loop, but decreases in the C-terminus of helix 3 (215–221; Figs. 6 and 7). In a number of WT PrP crystal structures the sidechain of R164 forms a pair of hydrogen bonds with the carboxyl group of E168 (2.5 and 3.1 Å in PDB 2W9E[29]). The introduction of the bulkier valine sidechain at residue 127 appears to sufficiently perturb the side-chain position of R164 such that the interactions with E168 are essentially removed (the equivalent distances are

3.2 and 3.8 Å; Fig. 8). Significantly, NMR chemical shift changes in the N$^ε$ signal from the R164 sidechain in V127/M129 PrP reflect this alteration in side-chain orientation, with the N$^ε$ being perturbed by the change in its proximity to the aromatic ring of Y128 and the change in hydrogen bonding of R164 N$^{H1}$. (Supplementary Fig. 6). The loss of these interactions is a likely source of the increase in the ms dynamics of both residues, which appears to be disseminated along the rest of the β2-α2 loop, as residue Q172, the other visible resonance in the β2-α2 loop, also displays a marked increase.

Similarly, the V129 polymorphism also affects ms dynamics, however, different residues are affected. V129 increases $R_{ex}$ values

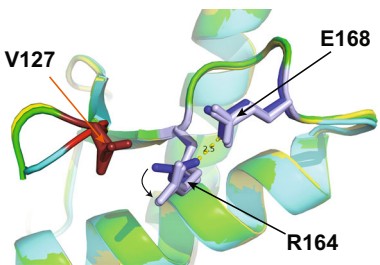

**Fig. 8 Perturbation of the R164 sidechain by V127 in V127 PrP crystals.** Comparison of residue 127, R164 and E168 side-chain positions in WT G127/M129 (cyan), V127/M129 (green) and V127/V129 PrP (yellow). Glycine 127 is coloured bright red, with the valine sidechains of residue 127 in V127/M129 and V127/V129 PrP coloured dark red. R164 and E168 in wild-type G127/M129 PrP are coloured dark blue, and lighter blue in both V127 variants. In both V127 variants the sidechain of R164 is sufficiently displaced from its position in the wild-type protein to significantly weaken the specific, strong (2.5 Å) hydrogen bonding interaction with E168 observed in wild-type G127/M129 PrP.

for G131 and itself (Fig. 6c, Supplementary Figs. 10 and 11). This alteration in G131 exchange dynamics has recently been observed in mouse PrP[42]. In addition, we also observe that the $R_{ex}$ values of D178 are markedly reduced in the V129 polymorph. This is illustrated by the reduction of line-broadening of D178 observed in the HSQC spectra of G127/V129 PrP (Supplementary Fig. 13).

This is notable as the residue 129 M/V polymorphism affects the disease phenotype of the pathogenic D178N mutation which causes inherited prion disease. D178N is associated with the clinico-pathological phenotype Fatal Familial Insomnia (FFI) when residue 129 is methionine, and CJD when it is valine[37]. In V127/M129 PrP the D178 HSQC resonance cannot be observed directly as it is heavily overlapped with that of V127, but an analysis of the intensity of signals in V127/M129 3D HNCO NMR spectra indicates that D178 does indeed experience ms dynamics, to a similar extent as wild-type G127/M129 PrP. This suggests that the V129 polymorphism alters PrP conformational variability independently of the 127 mutation.

A number of residues at the C-terminus of helix 3 (I215, Y218, E219, R220, E221 and S222) experience altered ms dynamics in V127/M129 compared with G127/V129 PrP. Residues I215/Y218/E221/S222 are on a face of the helix that interacts with residues in the β2-α2 loop (Fig. 7, Supplementary Fig. 11). For example, residues Y218 and S222 closely interact with M166, while I215 and Y218 interact with Q172. In particular, residues Y218 and E221 are subject to marked increases in $R_{ex}$ values in G127/V129 PrP, whereas in V127/M129 PrP there is a reduction at the C-terminus.

**V127 polymorphism does not perturb PrP stability**. To test whether the variations in dynamics have a substantial effect on local stability, hydrogen/deuterium exchange rates were obtained on V127/M129 PrP. The observed rates of hydrogen/deuterium exchange allow the determination of amide protection factors, which indicate the extent to which hydrogen bonding and burial prevents solvent access. The hydrogen/deuterium exchange data indicate that the protection factors and stabilities of the PrP secondary structure elements are indistinguishable between V127/M129 and G127/M129 PrP (Supplementary Fig. 14)[22,43]. The protection factors of the secondary structure elements, with the notable exception of the first strand of the β-sheet and in the vicinity of the disulphide bond, reflect the equilibrium constant between native and unfolded states of the protein ($K_{F/U}$)[43]. The majority of residues that display observable protection factors

therefore exchange from the globally unfolded state. This is also the case with G127/V129 PrP[22]. The V127 and V129 polymorphisms thus do not induce any alternatively folded states in which the core of the protein is destabilised. This is noteworthy as the first strand of the β-sheet (where the residue 127 and 129 polymorphisms lie) displays anomalously low protection factors corresponding to reduced stability (~30 times less than the other secondary structure elements), suggesting that its stability might be affected more readily by the protective polymorphisms. Notably, a number of regions that are subject to the ms conformational dynamics affected by both protective polymorphisms are in areas that do not display measureable hydrogen protection, for example the β2-α2 loop and the C-terminus of helix 3.

**Effect of V127 polymorphism on PrP in vitro fibrilisation**. The lack of major structural perturbation or altered stability of the V127 variant in comparison to WT PrP$^C$ suggests that the polymorphism may act by primarily affecting the efficiency of conversion of PrP$^C$ to its disease-associated aggregated form. To assess this we firstly examined the ability of the V127 variant to fibrilise under partially denaturing conditions. When agitated in 2 M GuHCl, PrP can be induced to form amyloid. Binding of the fluorescent thiazole dye thioflavin T to these β-sheet-rich fibrillar structures reports their formation, allowing a quantitative analysis of the kinetics of fibril formation[44]. We found that although V127/M129 PrP can be induced to fibrilise within the time scale of the experiment, it did so with a significantly longer lag-time than WT G127/M129 PrP (Fig. 9). This is particularly interesting as substitutions to valine, and other bulky hydrophobic residues typically promote β-sheet formation and self-association required for amyloid formation[45]. These data are however consistent with previously published data which modelled the effect of the V127 mutation on a mouse PrP background, and which indicated that the V127 variant is inherently more resistant to fibrilisation than WT PrP[46].

**Amplification of protease-resistant PrPSc seed (PMCA)**. Although fibrillar material can be generated using these partially denaturing conditions, the material generated in such reactions has not been shown to be reliably infectious. In contrast, the protein misfolding cyclic amplification (PMCA) technique[47] has been shown to amplify infectious and PK-resistant material with high fidelity. PMCA is a cyclical process where periods of conversion of substrate PrP$^C$ by small amounts of PrP$^{Sc}$ "seed" are interspersed with bursts of sonication. We performed PMCA reactions using brain homogenate from mice overexpressing WT G127/M129 PrP (Tg35) WT G127/V129 PrP (Tg152) or V127/M129 PrP (Tg183). Both WT PrP substrates allowed amplification of PK-resistant material. In contrast there was no amplification using V127/M129 as substrate (Fig. 10). These results are consistent with the observed disease characteristics of the vCJD strain type which propagates most readily with WT PrP with methionine at residue 129, and which failed to generate protease-resistant PrP or cause disease in transgenic mice expressing solely V127/M129 PrP[25].

## Discussion

This structural and biophysical study was stimulated by the remarkable effect of the V127 polymorphism on human prion propagation. Transgenic mouse transmissions show that V127 PrP is incapable of supporting prion transmission and propagation, consistent with the human clinical resistance data[24], and is even able to inhibit heterologous propagation of wild-type protein containing glycine at residue 127[25]. This differs from the residue 129 polymorphism, where similar studies suggest the importance

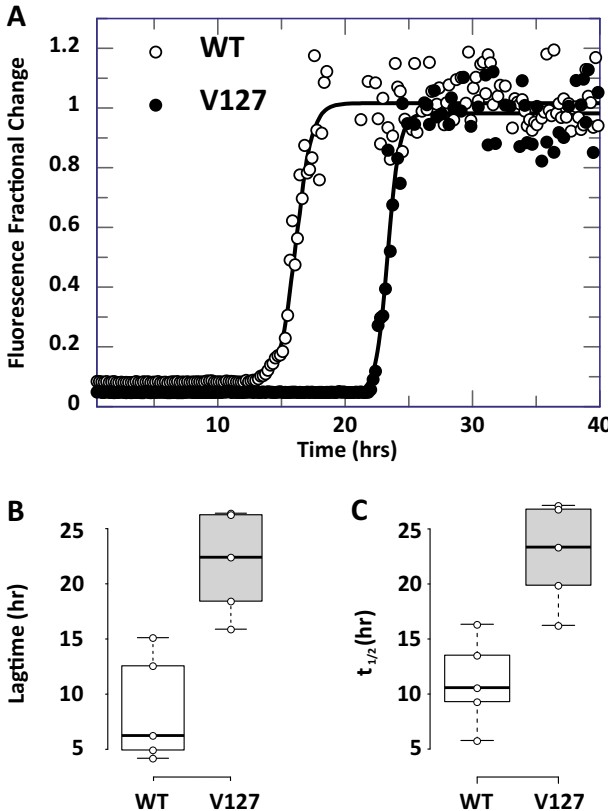

**Fig. 9 Quantitative analysis of the effect of V127 on PrP fibril formation.** **a** Formation of amyloid fibrils as reported by increasing Thioflavin T (ThT) fluorescence. The lines superimposed on the data are non-linear curve fits to Eq. (2), as described in the Methods section. **b**, **c** Fibrillogenesis of V127/M129 PrP occurs with significantly longer mean half- and lag times in comparison to WT G127/M129 PrP ($P \leq 0.01$, paired $t$-test). Centre lines show the medians for each data set; box limits indicate the 25th and 75th percentiles as determined by R software (http://shiny.chemgrid.org/boxplotr/); whiskers extend 1.5 times the interquartile range from the 25th and 75th percentiles, outliers are represented by dots. $N = 5$ sample points for each data set.

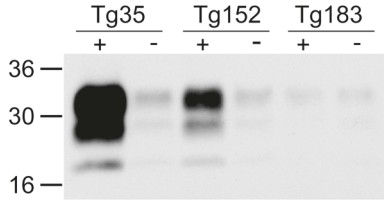

**Fig. 10 Western blot indicating the presence of PK-resistant material in PMCA reaction (+) and non-sonicated control (−) samples of PrP$^{Sc}$ amplified with Tg35 (huPrP G127/M129), Tg152 (huPrP G127/V129) and Tg183 (huPrP V127/M129) brain homogenate as substrate.** In each case, a small amount of seed can be detected in the non-sonicated control samples with varying levels of amplification observed in the reactions with different substrates. Of note is the lack of amplification with V127/M129 PrP (Tg183) as substrate.

of homologous protein interactions in prion propagation[17,18,20,48], and the preferential selection of different prion strains by PrP molecules with different primary structure as a result of conformational selection[2,16]. Unlike the residue 129 polymorphism, no strain switching or strain mutation was observed with V127, even in the homozygous state, indicating

that it confers complete resistance via the variant protein itself[25], leading to the hypothesis that an altered PrP$^C$ fold may be the cause of resistance to prion disease[34]. The profound effects that these mutations have on human prion disease pathogenesis may provide key insights into the mechanism of prion conversion, and have a wider relevance to other templated protein mis-folding diseases, where changing a single amino acid could have a similar dramatic effect, with potential significance for therapeutic strategies[9,10]. The structural consequences of the glycine to valine substitution at residue 127 on PrP$^C$ is therefore of major and wide potential interest.

Here, we have shown that there is a close similarity in overall structure between both V127 variants studied (V127/M129 and V127/V129), and wild-type G127/M129 PrP. Solution spectra (CD and NMR) confirm that the crystal structures faithfully reflect the solution structures of the proteins, allowing detailed analysis of the structural effect of the V127 polymorphism on PrP$^C$. We find little evidence for a major structural change in the β-sheet, or α-helices, in contrast to a recent NMR structure which identified unique features caused by the V127 polymorphism[34]. In particular we do not observe any displacement of amino acid side chains within the β-sheet, which are well defined by the electron density, and note that this crystal structure satisfies the inter-residue β-sheet NOE distance constraints used for the NMR structural study within 0.25 Å[34,49], apart from two which are satisfied within 0.43 Å and 1.03 Å (both to the Hε of Y162).

However, distinct perturbations of key regions which affect prion transmission and propagation are observed. Specifically, the V127 substitution reduces the conformational variability of the protein backbone immediately preceding the first strand of the β-sheet and radically alters the local backbone conformation. This facilitates the formation of an additional two intermolecular hydrogen bonds, which stabilise the native-state dimer association observed in the crystals. This dimeric association has been observed in a number of different PrP structures crystallised in the absence of antibody[30–32] (Fig. 4). In these, the PrP β-sheet interface is composed of two intermolecular hydrogen bonds, as in WT G127/M129 PrP[29]. The V127 dimer interface presented here is unique in both the length of the β-sheet interface and number of hydrogen bonds (4), and argues against the proposal that V127 disfavours native-state dimerisation, by reducing main-chain hydrogen-bond interactions[33,34].

Formation of the intermolecular β-sheet has been proposed as a possible initiation point for β-sheet-mediated oligomerisation to explain the genetic susceptibility and prion strain selection determined by the polymorphic residue 129 in human prion disease[23,29,30,32]. If β-strand interactions in this region of the protein mediate PrP interactions during PrP$^{Sc}$ formation, then the packing and geometry of this segment of the chain would have a strong selective effect on conformation, and also productive prion propagation[29]. No displacement of the protein backbone or stabilisation of the PrP$^C$ dimer interface is caused by the residue 129 polymorphism[22,32], which may explain the marked effect of the V127 polymorphism on prion pathogenesis.

The PrP$^C$ dimer interaction does not appear to be a crystal packing artefact as endogenous PrP$^C$ dimers have been detected in N2a cells and purified brain fractions[50,51] with the dimerisation region mapped to the hydrophobic domain of PrP (residues 112–133)[52]. PrP$^C$ dimerisation inhibits PrP$^{Sc}$ accumulation and prion replication[53,54], and has a dominant-negative inhibitory effect on the conversion of monomeric PrP$^C$[54]. These findings suggest that it may be possible to halt prion formation by stabilising PrP$^C$ dimers. Given the strengthened dimer association seen in the V127 crystals this may be one aspect of the protective mechanism of the V127 mutant. The conformational restriction imposed by the V127 polymorphism may also be sufficient to

inhibit homotypic protein–protein contacts in heterodimers of V127 and G127 PrP, or prevent the formation of extended β-sheet structure required to convert the PrP N-terminal unstructured region into protease-resistant β-enriched forms[41,42,46,55].

Alternatively, the marked alteration in local backbone conformation and increased stability of intermolecular β-sheet interactions may prevent PrP folding into a thermodynamically permissible prion assembly[3,4,56,57]. As V127 PrP also inhibits the generation of infectious assemblies of wild-type PrP, this would suggest that it must be either capping nascent prion assemblies or structurally weakening infectious prion assemblies on incorporation. The PMCA data presented here indicates that V127 PrP is not a permissive substrate for amplification of protease-resistant PrP[Sc] disease seed. It is possible that the polymorphism introduces a protease cleavage site, which would lead to the destruction of a polymer formed of the homomeric V127 protein. PrP V127 incorporation would also dope, with a dose-dependent effect, heteromeric polymers composed of both G127 and V127 PrP variants. Their reduced stability could increase cellular clearance, which would also explain the "dominant negative" effect of V127 on prion propagation[58].

In addition to these local structural perturbations, long-range sequence interactions between the protective residue 127 and 129 polymorphisms affect the conformational distribution of a spatially distinct region including the β-sheet, β2-α2 loop, the C-terminus of helix 3 and the disease-associated residue D178[37]. These conformationally variable elements have been shown to be key determinants of prion transmission and cross-species prion susceptibility, and a number are associated with inherited forms of prion disease, for example G131V, D167N, V210I, E211Q, Q212P and Q217R[26,28,37,59–64]. In particular, the β2-α2 loop (residues 165–172) has been proposed to be a key modulator of prion transmission and disease-associated PrP misfolding[26–28,35,36,58]. V127 alters the structural flexibility of the β-sheet and conformational dynamics of the β2-α2 loop by disrupting the electrostatic interaction between R164 of the β-sheet and E168 within the adjacent β2-α2 loop. Loss of this interaction would disrupt hydrogen bonding and close packing between Y169, F175 and D178 and destabilise the dominant $3_{10}$–helical conformation of the β2-α2 loop[27,36,40] (Supplementary Figs. 11 and 12). Notably, residue 168 (human numbering) is polymorphic in sheep PrP, in which either glutamine or arginine can be accommodated[65]. The arginine polymorph, which would also diminish the electrostatic interaction with R164, is associated with resistance to scrapie, and potently inhibits prion conversion[60,62]. These observations support the notion that increased conformational variability associated with the loss of charge interaction between R164 and E168 increases resistance to prion disease. Substitution of arginine for glutamine at the totally conserved Q172, the dynamics of which are significantly altered in the V127 polymorph also potently inhibits in vitro prion infectivity[66].

The recent NMR structural study of V127 PrP also identified significant alterations in the conformational dynamics in the β-sheet and α-helix 2 as a result of the polymorphism[34]. Although there are a number of residues perturbed in both studies, the marked effect that we observe on the dynamics of the β2-α2 loop was not observed in the previous study. The differences in protein dynamics may be ascribed to the different solution conditions for data acquisition. In this study PrP was crystallised at pH 8.0, with the NMR solution dynamics acquired at pH 5.5, whereas the previous NMR study used pH 4.5, which may cause glutamate and aspartate side chains to adopt an unphysiological protonated state[34]. The calculated p$K_a$'s of E168 and D167 in V127/M129 PrP are 4.7 and 4.4, respectively[67]. The hydrogen-bonding and charge interactions of the β2-α2 loop involving the side-chains of these residues will be weakened through increased protonation.

This will likely cause altered conformational dynamics in the β2-α2 loop and adjacent regions. For example, the lack of assigned resonances for R164 by Zheng et al.[34] may be ascribed to intermediate timescale exchange broadening these signals beyond detection. Given the p$K_a$'s of interacting residues, the pH at which structural studies are carried out might thus be very significant.

It is of great interest that both the V127 and V129 polymorphisms have long-range effects on the conformational distribution of these regions of the protein and also the β-sheet. The correlation between these conformationally variable regions of PrP and its propensity to form disease-related isoforms suggest that these regions of the protein are important in prion assembly-PrP[C] interactions, determining efficient binding and conversion[27,36]. Indeed, relatively subtle variations in the strength and orientation of monomer docking can dramatically affect the productivity of fibrillogenic interactions and determine barriers to amyloid formation[68]. Given the effect of the V127 polymorphism on the PrP β-sheet backbone geometry and intermolecular association of PrP monomers observed here, it is tempting to speculate that dimerisation via the formation of the intermolecular PrP β-sheet may be a critical event in PrP oligomerisation and prion propagation and thus explain the exceptional effect of the residue 127 polymorphism on human prion disease.

## Methods

**Recombinant PrP and antibodies.** Recombinant human PrP containing residues 119–231 (PrP[119–231]) was produced and purified as previously described[69]. This length of construct was chosen as the PrP N-terminus up to approximately residue 125 is unstructured in full-length (residues 23–231) and truncated (residues 91–231) PrP and compromises the NMR dynamics characterisation due to its effect on the rotational tumbling of the structured domain[70,71]. Removal of the N-terminal tail does not affect the structure or local structural fluctuations of the PrP structured globular domain[72,73]. PrP containing valine at residue 127 (V127/M129 and V127/V129), and wild-type PrP with glycine at residue 127 on both 129 methionine and valine backgrounds (G127/M129 and G127/V129) were expressed and purified for biophysical analysis. ICSM 18 was purchased from D-Gen Limited. The Fab fragment of ICSM 18 was prepared by limited papain digest of the mature antibody followed by purification using gel filtration chromatography.

**Crystallisation conditions.** ICSM 18-Fab and PrP were mixed at a molar ratio of 3:1 for preparation of the complex prior to crystallisation, and incubated at room temperature for 30 min before buffer-exchanging the complex into 50 mM Tris, 150 mM NaCl, pH 8.0 and filtering through a 0.22 μm membrane prior to crystallisation. Crystals of the complex were obtained by using the sitting-drop vapour diffusion technique; droplets containing 5–6 mg/mL PrP in 0.4 M and 0.75 M ammonium sulphate, 0.05 M Tris (pH 7.5 and 8.0) were equilibrated over wells containing 0.8 M and 1.5 M ammonium sulphate, 0.1 M Tris (pH 7.5 and 8.0). Crystals, round in shape, grew in 6 months to 0.05–0.2 mm diameter.

**In situ data collection and analysis.** Data were collected at room temperature in situ on beamline I03 at Diamond Light Source, with the crystallisation plates sealed in biohazard bags. Multiple wedges of data were collected from different parts of the same crystal, and from different crystals, and scaled together to provide a complete dataset. We typically collected 15° of data from each crystal in 0.3° oscillations. Data were integrated with *XDS*[74] and then subsequently *BLEND*[75] was used to analyse how well the different wedges of data scaled together and the results used to decide which datasets should be scaled and merged with *AIMLESS*[76].

**Structure determination and refinement.** The structures were solved by molecular replacement using PHASER[77] with the heavy and light chains of the Fab fragment of antibody ICSM18 and the PrP molecule used as search models (protein databank accession code 2W9E). Electron density maps were inspected and the models built using *COOT*[78] followed by refinement with *REFMAC5*[79]. Data collection and final refinement statistics are summarised in Supplementary Table 1. Ramachandran statistics for V127/M129 and V127/V129 structures (in parentheses) are as follows; residues in most favoured region: 96.7% (96.7%), residues in additionally allowed regions: 3.3% (3.3%) and residues in disallowed regions: 0.0% (0.0%)[80]. The final coordinates of the V127/M129 and V127/V129 structures have been deposited in the Brookhaven Protein Data Bank (http://www.rcsb.org), with accession numbers 6SV2 and 6SUZ respectively.

**NMR sample preparation and spectroscopy.** For the NMR study [15]N & [13]C/[15]N-labelled samples of PrP were prepared. Following purification, protein samples

were either (A) buffer-exchanged into 20 mM sodium acetate, containing 1.5 mM sodium azide ($NaN_3$), pH 5.5 through dialysis, then concentrated in Vivaspin 20 centrifugal concentrators to protein concentrations of 0.8–1.2 mM or (B) dialysed against deionised water then lyophilised and resuspended into 20 mM sodium acetate, 1.5 mM $NaN_3$, pH 5.5. 10% $D_2O$ (v/v) was added to the NMR samples to provide the lock signal, together with TSP as the chemical shift reference to 1 mM final concentration. NMR samples were placed in Sigma FEP NMR sample tube liners (Z286397–1EA), held within Wilmad PP-528 NMR tubes for NMR data acquisition.

Assignment spectra for V127/M129 PrP were acquired at 303 K on Bruker DRX-600 and DRX-800 spectrometers, with [15]N-relaxation measurements for V127 and V129 PrP acquired at 298 K on Bruker Avance III 500 and 800 MHz spectrometers, all equipped with 5 mm [13]C/[15]N/[1]H triple-resonance probes. Sensitivity-enhanced [1]H-[15]N HSQC[81,82] and standard triple-resonance experiments[83] with uniformly [13]C/[15]N-labelled protein (HNCA, HNCACB, CBCA (CO)NH and HNCO) were used to obtain V127/M129 backbone resonance assignments. Proton chemical shifts were referenced to TSP. [13]C & [15]N chemical shifts were calculated relative to TSP, using the gyromagnetic ratios of [13]C, [15]N and [1]H ([15]N/[1]H = 0.101329118; [13]C/[1]H = 0.251449530). For residue 127, when comparing [13]C chemical shifts, the difference in residue type was compensated for by subtracting residue-specific random coil shifts (glycine/valine) to generate secondary chemical shifts, which were then subtracted[84]. NMR data were processed and analysed using Felix 2007 (Accelrys, San Diego), Topspin (v 3.2, Bruker) and CCPN Analysis (v. 2.3.1)[85] software.

**Spin relaxation measurements**. Spin relaxation measurements ($T_1$, $T_2$ and [15]N ([1]H)-NOE) were acquired on 1 mM [15]N-labelled $PrP^{119–231}$ WT (G127/M129) PrP, V127 (V127/M129) PrP and V129 (G127/V129) PrP as described in Yip et al.[86]. Briefly, by using this methodology, heating compensation was improved by the incorporation of a compensation block based on the relaxation block, followed by a pre-scan [1]H saturation sequence and constant length recovery period. The $T_1$ data were obtained using [15]N relaxation delays of 50, 100, 200*, 300, 500, 800*, 1000, 1500, 2000, 3000, 4000 and 5000 ms. The $T_2$ data were obtained using [15]N relaxation delays of 8.5, 17.0, 33.9*, 50.9, 67.8, 101.8*, 135.7, 186.6, 254.4 ms (500 MHz) and 7.8, 15.7, 31.4*, 47.0, 62.7*, 94.1, 125.4, 172.5 and 235.2 ms (800 MHz; the asterisks denote duplicate measurements). $T_1$ and $T_2$ datasets were recorded as pseudo 3D experiments, with randomised order of time increments. Two separate nitrogen offsets were used to reduce build-up of off-resonance artefacts during the CPMG block of the $T_2$ measurements. For the [15]N-[1]H-NOE measurement, two two-dimensional spectra were acquired with a relaxation delay of 6 s between scans. Spectra were collected with $t_1$ acquisition times of 94.7 ms (500 MHz)/59.2 ms (800 MHz) and $t_2$ (direct) acquisition times of 127.8 ms (500 MHz)/91.8 ms (800 MHz). Errors for time-series $T_1$ and $T_2$ data were calculated from the overall standard deviation for duplicate data points in the series. Errors for the NOE data were estimated from measurements of the root mean-square deviation of the base-plane noise in those spectra. Non-linear least-squares (Levenberg-Marquardt) fitting of two-parameter exponential functions to decay data was performed using in-house routines using Numerical Python.

**Modelfree analysis**. Protein dynamics were analysed by Relax (v. 3.3.1)[38,39], using the $T_1$, $T_2$ and [15]N{[1]H}-NOE spin relaxation data. Reduced spectral density mapping analysis, as implemented by the Relax default $J_{(w)}$ mapping script mode was used to obtain $J_{(\omega)}$ values for each given field strength. A full Modelfree analysis[87] was carried out using the "d'Auvergne" protocol within Relax. Extended order parameters ($S_2$, $S^f_2$, $S^s_2$), the effective correlation time for fast internal motions ($t_e$) and intermediate exchange broadening contribution ($R_{ex}$) values were obtained using this protocol.

**Amide exchange protection experiments**. Hydrogen-deuterium exchange rates ($k_{ex}$) were determined by adding 260 μl 20 mM sodium acetate, 1 mM sodium azide, pH 4.5, dissolved in 100% (v/v) $D_2O$ to lyophilised PrP samples, to obtain final protein concentrations of 1 mM. A series of sensitivity-enhanced [1]H-[15]N HSQC spectra[81,82] were acquired at 293 K on a Bruker DRX-800 spectrometer. The decay curves of the [1]H-[15]N HSQC cross-peaks were fitted to single exponential decays with offset, and protection factors ($k_{ex}/k_{int}$) for observable amides were determined using intrinsic amide exchange rates[88] ($k_{int}$). Acquisition of the first experiment began ~5 min after mixing, setting a lower limit on the detection of protection factors of ~5.

**Circular dichroism**. Circular dichroism was measured at 25 °C with a Jasco J-715 spectropolarimeter, using a 0.1 cm pathlength quartz cuvette. The sample temperature was controlled with a circulating water bath. Far-UV (amide) CD spectra were recorded between 180 nm and 300 nm with 20 μM protein (2 nm bandwidth; Data Pitch 0.5 nm). In all, 10–50 spectra were averaged.

**Equilibrium unfolding measurements**. For equilibrium unfolding experiments, 6 μM protein was incubated in 10 mM HEPES, 25 mM NaCl pH 7.5, and increasing concentrations of GuHCl denaturant. Molecular ellipticity ([θ], degree $M^{-1}$ $cm^{-1}$)

was recorded at 222 nm (5 nm bandwidth; 20 s integration time). The denaturation profile for each protein was measured in three separate experiments.

**Conversion to molar denaturant activity**. To allow more accurate extrapolation of data to calculate folding parameters in the absence of denaturant and the free energy change of protein folding ($\Delta G$), denaturant concentration ([GuHCl]) was converted to molar denaturant activity ($D$), as described in Parker et al.[89], using $C_{0.5} = 7.5$.

**Equilibrium constant between folded and unfolded states**. For the two-state equilibrium unfolding transitions, data were fitted to the following equation, where $K$ and $K_{(W)}$ are equilibrium constants between the folded and unfolded states at a given denaturant activity ($D$) and in water, respectively, and $m$ describes the sensitivity of the equilibrium to denaturant activity[89].

$$K = K_{(W)} \exp(m.D) \qquad (1)$$

For visual representation of the data shown, data were converted to proportion folded, $\alpha_F$, using the following, $\alpha_F = (K/(1 + K))$. Data fitting was carried out using GraFit (Erithacus software). The significance of the differences in free energy for folding and $m$ values between the three variants characterised were determined by paired two-tailed Student's $t$ test.

**Quantitative analysis of the kinetics of PrP fibril formation**. Recombinant V127/M129 and WT G127/M129 PrP (residues 119–231) was dialysed into 20 mM sodium acetate, 2 mM sodium azide, pH 6.0, and then denatured by the addition of GuHCl to a final concentration of 6 M. Denatured PrP was then diluted to a final concentration of 10 μM in 20 mM sodium acetate, 2 M GuHCl, 10 mM EDTA, 100 μM Thioflavin T (ThT) and pH 6.0. All solutions were filtered through a 0.22-μm filter to remove particulates. In all, 200 μl aliquots were placed in silanised Greiner 96-well flat-bottomed plates (#655077) containing four 0.5-mm diameter zirconium ceramic beads in each well to assist agitation. The plates were incubated at 37 °C with constant agitation in a Tecan Infinite F200 Microplate Fluorimeter. Fibril formation was monitored through the increase in ThT fluorescence (excitation 430 nm, emission 485 nm), with readings acquired every 600 s. Five replicates were used for each PrP sample.

To determine the half- and lag-times for fibril formation, data were fitted to an empirical function described by Nielsen et al.[90].

$$\text{Fi} + \text{Ff}/\{1 + \exp[-(t - t_m)/\tau]\} \qquad (2)$$

where Fi is the initial fluorescence reading, Ff is the final fluorescence reading, $t$ is time, $t_m$ is the time taken to half maximal fluorescence and $\tau$ is the reciprocal of the propagation rate during the rise phase [$1/k_{(apparent)}$]. Lag-time is defined as $t_m - 2\tau$.

**Formation of protease-resistant PrP by PMCA amplification**. PMCA substrate homogenates were prepared from mice that had been perfused with PBS containing 5 mM EDTA at the time of death. PrP-null ($Prnp^{o/o}$), Tg35 (homozygous for huPrP G127/M129), Tg152 (homozygous for huPrP G127/V129) or Tg183 (homozygous for huPrP V127/M129)) mouse brains[25] were homogenised in cold conversion buffer (PBS containing 150 mM NaCl, 1.0% (v/v) Triton X-100, 4 mM EDTA and 1× Complete protease inhibitor (Roche Applied Science)), using a Duall tissue grinder to give a 10% (w/v) homogenate.

Substrates were seeded with a 1/100 dilution of vCJD (I4618) 10% brain homogenate in PBS. Each reaction mixture was divided in two prior to PMCA with one half stored at −70 °C as a minus PMCA control. PMCA consisted of 96 cycles of 30 s sonication every 30 min in a Misonix S3000 at 75% power output (Misonix, Farmingdale, NY), reactions were carried out with 40 μl substrate in 200-μl thin-walled PCR tubes at 35 °C.

Samples were digested with 50 μg $ml^{-1}$ proteinase K (PK) for 1 h at 37 °C. The reaction was stopped by the addition of AEBSF in SDS-loading buffer and samples were boiled for 10 min before running on 16% Tris-glycine gels. Western blotting was carried out according to Unit protocol, using 3F4 (Merck Inc, N.J., U.S.A) as the primary antibody and goat anti-mouse IgG conjugated to alkaline phosphatase (Sigma A2179) as the secondary antibody.

**Statistics and reproducibility**. In the reported experiments, each protein sample was identically-engineered. The sample size ($n$) of each experiment is provided in the corresponding figure captions in the main manuscript and supplementary information files. Sample sizes were chosen to support meaningful conclusions. All in vitro folding experiments were replicated at least three times. In vitro fibrillisation assays were replicated five times. $T_1$ and $T_2$ NMR data were recorded with randomised order of time increment and each included one duplicate dataset. Replicate experiments were successful. Investigators were not blinded during experimental measurements or data analysis.

**Reporting summary**. Further information on research design is available in the Nature Research Reporting Summary linked to this article.

## Data availability

The atomic coordinates for the crystal structures described in this paper have been deposited in the Brookhaven Protein Data Bank (https://www.rcsb.org/) (accession nos. 6SV2 (V127/M129 PrP) & 6SUZ (V127/V129 PrP)). The data that support the findings of this study are available from the corresponding author upon reasonable request.

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

## Acknowledgements

We thank Richard Newton for the preparation of figures. This work was supported by the Medical Research Council. We are grateful to the staff at Diamond Light Source for access to X-Ray diffraction data collection facilities. We gratefully acknowledge the longstanding and major contribution of the late Anthony Clarke to our structural studies.

## Author contributions

L.L.P.H., R.C., D.S., M.J.C., A.M.H., K.M., R.L.B., G.S.J., J.P.W. and J.C. designed research; L.L.P.H., R.C., D.S., M.B., E.B.S., S.F., M.J.C., A.M.H., K.M., R.L.B. and J.P.W. performed research; D.S. and K.M. contributed new analytic tools; L.L.P.H., R.C., M.J.C., A.M.H., K.M., R.L.B., J.P.W. and J.C. analysed data; L.L.P.H., R.C., K.M., R.L.B., J.B., J.P.W. and J.C. wrote the paper.

## Competing interests

G.S.J. and J.C. are shareholders, and J.C. a director of D-Gen Limited, an academic spin-out company in the field of prion disease diagnosis, decontamination, and therapeutics, which provided the ICSM18 monoclonal antibody used in this study. The remaining authors share no competing interests.
