## [Peer Review File · Communications Biology]

Reviewers' comments:

Reviewer #1 (Remarks to the Author):

In the manuscript "Structural effects of the highly protective V127 polymorphism on human prion protein" the Authors showed that the V127 mutation imposes local conformational changes to some of the residues and induces conformational restrictions in this region of the protein using different structural and biophysical methods. Though the manuscript provides a strong structural aspect for the mechanistic explanation for the protective effect of V127 mutant I have the following comments and concerns about the manuscript. It might be accepted after suggested modifications.

1. Some of the references are either missing or not cited with proper statements in the introduction such as: Page 1, para 2, sentence one; Page 1, para 3, sentence one.
2. There are some abbreviations that are explained in the whole manuscript but used several times such as PRNP.
3. If human PrP is crystallizable in the absence of antibodies, then what is the rationale here to use an antibody to increase complications in the study? Authors might need to crystallize them in the absence of antibodies to check antibody driven conformation changes in the mutant which will be more physiologically relevant.
4. Since the prions are beta-sheet rich aggregated material and Thioflavin T and Congo Red are known to bind to this beta rich aggregated material. Authors need to show some of the biochemical data showing the binding of Thioflavin T or Congo Red to show the effect of the V127 polymorphism.
5. Figure 1: To make the figure better understandable, I would suggest changing colors of the two models so that they are better visible. Currently, the colors are not standing out properly.
6. Figure 1: figure legend: the model representation is not ribbon; it is a cartoon. The authors also did not mention which program, they used to prepare these figures.
7. Figure 2: this figure does not provide much information. The residues which experience altered ms dynamics can be shown in the table. Authors may compare other structures such as human PrP without antibody and any other polymorphism structure already available in the PDB.
8. Figure 5: figure legend: either authors forgot to label the E219 or mislabeled it with Y218. It needs to be corrected. Also, authors need to expand the top part of the figure for better illustration. H-bonds (Y169/D178) are not labeled properly.

Reviewer #2 (Remarks to the Author):

This is a carefully carried out, well-presented study using both crystal structures, NMR dynamics and H/D protection factors and represents a large body of work. It is suitable for Communications Biology. It describes the cellular-PrP G127V mutation which has a protective effect on the transmission of Prion disease.

Key Concerns

(1) Throughout the manuscript it is assumed that the G127V mutant is protective because of its impact on the structure of PrP^{cellular} isoform (line 294). The mutation may have little or nothing to do with its impact on the confirmation of the cellular form, but rather the impact on a partially or completely unfolded/misfolded PrP intermediate structure. Thus the mutation may actually impact a largely or completely unfolded PrP intermediate structure rather than the cellular structure studied. If this is the case studies of the PrP^{cellular}(G127V) structure will have little relevance to the ability or otherwise of PrP assembling into amyloid fibrils/oligomers.

(2) There is a paper published in scientific reports from last year (Zheng, Zhang, Lin 2018 Ref29) that has a significant overlap with this study the publication shows quite a careful NMR dynamics and molecular dynamics study of the same G127V mutant.

Line 364,- As Ref 29 is important- need to be specific about the differences in the experimental conditions.

(3) Dimer structures in X-ray crystals can be artifacts of high concentrations used in crystallization and may not occur in vivo. I appreciate the NMR studies show changes in dynamics. However this is for monomeric PrPC ? does not have dimer Hydrogen-bonding? have there been any attempt to determine the infinity of the dimer, to indicate this could happen in vivo.

(4) An important aspect of this protective mutant is its ability or otherwise to form amyloid fibrils/oligomers in vitro (as well as vivo). I am surprised the authors have not discussed ThT kinetic studies, are there any published? If not, why have the authors not performed these standard experiments themselves. It would support (or otherwise) the direct biophysical impact of the mutation on amyloid formation. (I note for example that the protective mutation found in amyloid beta peptide does not markedly inhibit fibril formation of A β in vitro).

(5) Line 38 of abstract, is there much evidence that V127 is a key region associated with the development of prion disease. More than 30 point mutations have been link with inherited prion disease, these are from all parts of the Prion primary sequence, not just near V127. Furthermore, PrPSc core beta-sheet structure is in large parts of the sequence not just near 127.

(6) the reader is left feeling that the studies of the PrPC structure and dynamics do not clearly identify the reasons for the protective nature of the G127V mutation. See for example the vagueness of the conclusion within the abstract. Can this be made firmer

Other minor suggestions

(7) Substitutions to Valine (a large bulky hydrophobic residue) will typically promote beta-sheet and self-association required for amyloid formation There are numerous examples of this in familial prion disease GSS including P102L, P105L, A117V and G131V all of which mutate to V/L and promote prion disease. I believe this is worth a mention as it further highlights G127V is an interesting exception.

(8) Line 75, - Ref typo

(9) Line 79, 2009 not recently

(10) Line 162, posit = suggest

(11) Line 331, needs a reference

(12) Line 372 Ref typo

Dr. Anam Akhtar,
Associate Editor,
Communications Biology

Response to reviewer and editorial comments for manuscript COMMSBIO-19-1502-T (Structural effects of the highly protective V127 polymorphism on human prion protein):

Dear Dr. Akhtar,

Many thanks for forwarding the reviewer and editorial comments which we found extremely useful. We have incorporated these into a revised manuscript which we believe is distinctly improved. Given below are our detailed responses to each query. The original reviewer/editorial comments are shown in italics, and changes to the manuscript are shown in bold. We have also highlighted all changes in a revised manuscript text file.

Specific reviewer comments:

Reviewer #1

In the manuscript "Structural effects of the highly protective V127 polymorphism on human prion protein" the Authors showed that the V127 mutation imposes local conformational changes to some of the residues and induces conformational restrictions in this region of the protein using different structural and biophysical methods. Though the manuscript provides a strong structural aspect for the mechanistic explanation for the protective effect of V127 mutant I have the following comments and concerns about the manuscript. It might be accepted after suggested modifications.

- 1. Some of the references are either missing or not cited with proper statements in the introduction such as: Page 1, para 2, sentence one; Page 1, para 3, sentence one.*
 - We have provided additional references (e.g. Prusiner SB *PNAS* 1998 95, 13363; Bolton *et al.*, *Science* 1982 218, 1309; Meyer *et al.*, *PNAS* 1986 83, 2310; Khan *et al.*, *PNAS* 2010 107, 19808) which corroborate the statements made in the introduction and other sections of the manuscript.**
- 2. There are some abbreviations that are explained in the whole manuscript but used several times such as PRNP*
 - We have added the missing abbreviations to the abbreviations section.**
- 3. If human PrP is crystallizable in the absence of antibodies, then what is the rationale here to use an antibody to increase complications in the study? Authors might need to crystallize them in the absence of antibodies to check antibody driven conformation changes in the mutant which will be more physiologically relevant..*
 - Following your request we attempted crystallisation of V127 PrP in the absence of antibody. We were unsuccessful within the constraints of the revision of the current manuscript. We are confident, however, that our data set, in combination with a more thorough review of literature that is included in the revised manuscript, addresses the reviewer's concerns (new Figure 2). We have previously crystallised wild-type PrP in the absence of antibody under a number of different conditions, and co-crystallising V127 PrP with antibody generated high-quality well-diffracting crystals with the exactly the same space group and antibody-binding interaction as wild-type PrP. Indeed, the rmsd of backbone heavy atoms between wild-type and variant structures is less than 0.24 Å. The symmetry-related dimers observed in our PrP-antibody crystals are observed in many other PrP crystal structures, such as sheep, rabbit, and a number of human PrP mutants (Haire *et al.*, 2004; Khan *et al.*, 2010, Lee *et al.*, 2010). These were obtained in the absence of antibody, and in**

different crystallographic space groups. As the ICSM18 antibody epitope is remote from the PrP dimer interface, and there is a co-occurrence of β -sheet structure between PrP molecules in different apo-crystal structures, this suggests that the dimer interaction we observe is not an artefact of crystal packing, or the antibody complex, but instead might have a greater biological significance, especially given the powerful genetic susceptibility and prion strain selection determined by the polymorphic residues 127 & 129 in human prion disease. Our response to Reviewer 2's Point 3 also highlights the potential role of PrP dimerisation in prion propagation. We believe that we have the appropriate control for the V127 structure in the form of the wild-type PrP-antibody crystal structure, and that we are not observing antibody-driven conformational changes in the V127 crystal structure. **We have added these points to paragraph 3 of the discussion in the revised manuscript.**

4. *Since the prions are beta-sheet rich aggregated material and Thioflavin T and Congo Red are known to bind to this beta rich aggregated material. Authors need to show some of the biochemical data showing the binding of Thioflavin T or Congo Red to show the effect of the V127 polymorphism.*

- Both reviewers raise an important point regarding the impact of the V127 mutation on the ability of PrP to fibrillise. We have performed the biophysical experiments suggested, which show that the V127 mutation does indeed inhibit PrP fibrillation (new Figure S17). Our results are consistent with previously published data (albeit modelled on a mouse PrP background) (Sabareesan & Udgaonkar, *Biochemistry*, 2017). In addition, we also include PMCA data in the paper which demonstrates that V127 PrP is not a permissive substrate for amplification of PrP^{Sc} disease seed (new Figure S18). This technique more closely models prion replication and crucially generates authentic prion infectivity. These data are very informative on the direct biophysical impact of the mutation on amyloid formation, and are consistent with the transgenic mouse modelling study which showed that V127 expressing mice were completely resistant to prion infection. **We have added the above data to the results section, and considered its relevance in an amended discussion section.**

5. *Figure 1: To make the figure better understandable, I would suggest changing colors of the two models so that they are better visible. Currently, the colors are not standing out properly.*

- **We have changed the colours of Figure 1, which we believe better distinguishes the two models.**

6. *Figure 1: figure legend: the model representation is not ribbon; it is a cartoon. The authors also did not mention which program, they used to prepare these figures.*

- **We have corrected the Figure 1 legend, and also added a reference for the program used to prepare this figure and other structural figures in the manuscript.**

7. *Figure 2: this figure does not provide much information. The residues which experience altered ms dynamics can be shown in the table. Authors may compare other structures such as human PrP without antibody and any other polymorphism structure already available in the PDB.*

- We thank the reviewer for his suggestion. **We have relegated the original Figure 2 to Supplementary Information, and replaced it with a figure which compares our antibody-complexed PrP structure with other PrPs which have been crystallised in the absence of antibodies. We also make reference to the extended V127 dimer interface in relation to these other PrP dimers in the discussion section.**

8. *Figure 5: figure legend: either authors forgot to label the E219 or mislabeled it with Y218. It needs to be corrected. Also, authors need to expand the top part of the figure for better illustration. H-bonds (Y169/D178) are not labeled properly.*

- We thank the reviewer for spotting this typographical error. **We have corrected the figure legend. We have amended Figure 5 to include an expanded view of the top part of the original figure as a separate panel. This allows proper labelling of the Y169/D178 H-bond.**

Reviewer #2:

This is a carefully carried out, well-presented study using both crystals structures, NMR dynamics and H/D protection factors and represents a large body of work. It is suitable for Communications Biology. It describes the cellular-PrP G127V mutation which has a protective effect on the transmission of Prion disease.

Key Concerns:

1. *Throughout the manuscript it is assumed that the G127V mutant is protective because of its impact on the structure of PrP cellular isoform (line 294). The mutation may have little or nothing to do with its impact on the confirmation of the cellular form, but rather the impact on a partially or completely unfolded/misfolded PrP intermediate structure. Thus the mutation may actually impact a largely or completely unfolded PrP intermediate structure rather than the cellular structure studied. If this is the case studies of the PrP Cellular (G127V) structure will have little relevance to the ability or otherwise of PrP assembling into amyloid fibrils/oligomers.*

- We completely agree with the reviewer's point. Our focus in this paper is on the structural effect of the V127 mutation on PrP^C, however **we do consider in the discussion that the mutation may prevent PrP folding into a thermodynamically permissible prion.**

2. *There is a paper published in scientific reports from last year (Zheng, Zhang, Lin 2018 Ref29) that has a significant overlap with this study the publication shows quite a careful NMR dynamics and molecular dynamics study of the same G127V mutant. Line 364,- As Ref 29 is important- need to be specific about the differences in the experimental conditions.*

- We appreciate the reviewer's point and have added the following to the text:

The recent NMR structural study of V127 PrP also identified significant alterations in the conformational dynamics in the β -sheet and α -helix 2 as a result of the polymorphism (Zheng, 2018). Although there are a number of residues perturbed in both studies, the marked effect that we observe on the dynamics of the β 2- α 2 loop was not observed in the previous study. The differences in protein dynamics may be ascribed to the different solution conditions for data acquisition. In this study PrP was crystallised at pH 8.0, with the NMR solution dynamics acquired at pH 5.5, whereas the previous NMR study used pH 4.5 (Zheng, 2018), which may cause glutamate and aspartate side chains to adopt an unphysiological protonated state. The calculated pK_a's of E168 and D167 in V127/M129 PrP are 4.7 and 4.4, respectively (Anandakrishnan, 2012). The hydrogen-bonding and charge interactions of the β 2- α 2 loop involving the side-chains of these residues will be weakened through increased protonation. This will likely cause altered conformational dynamics in the β 2- α 2 loop and adjacent regions. For example, the lack of assigned resonances for R164 by Zheng et al may be ascribed to intermediate timescale exchange broadening these signals beyond detection (Zheng, 2018). Given the pK_a's of interacting residues, the pH at which structural studies are carried out might thus be very significant.

3. *Dimer structures in X-ray crystals can be artefacts of high concentrations used in crystallization and may not occur in vivo. I appreciate the NMR studies show changes in dynamics. However this is for monomeric PrP^C? does not have dimer Hydrogen-bonding? have there been any attempt to determine the infinity of the dimer, to indicate this could happen in vivo.*

- Several published studies show that dimerisation may have an important role in prion propagation *in vivo* (Priola *et al.*, (1995) *J. Biol. Chem.* **270**, 3299; Meyer *et al.*, (2000) *J. Biol. Chem.*, **275**, 38081; Rambold *et al.*, (2008) *EMBO J.* **27**, 1974; Engelke *et al.*, (2018) *J. Biol. Chem.*, **293**, 8020), indicating that the observed dimer is not a crystallisation artefact. The latter study in particular (Engelke *et al.*) showed that PrP^C dimerisation prevented its conversion into PrP^{Sc}, and that PrP^C dimers had a dominant-negative inhibitory effect on the conversion of monomeric PrP^C, which was also observed in the prion transmission studies to transgenic mice expressing both V127 and WT G127 PrP. These findings suggest that it may be possible to halt prion formation by stabilizing PrP^C dimers, and that this may be one aspect of how the V127 mutation prevents prion propagation, contrary to the conclusions of a previously published molecular dynamics simulation (Zhou *et al.*, *Sci. Rep.* (2016), **6**, 21804). **These points have been added to the manuscript discussion section.** We have not attempted to determine the affinity of the dimer association, as this would be beyond the scope of an already very extensive manuscript.
4. *An important aspect of this protective mutant is its ability or otherwise to form amyloid fibrils/oligomers in vitro (as well as vivo). I am surprised the authors have not discussed ThT kinetic studies, are there any published? If not, why have the authors not preformed these standard experiments themselves. It would support (or otherwise) the direct biophysical impact of the mutation on amyloid formation. (I note for example that the protective mutation found in amyloid beta peptide does not markedly inhibit fibril formation of Aβ in vitro).*
- This has been addressed above in response to Reviewer 1's point #4
5. *Line 38 of abstract, is there much evidence that V127 is a key region associated with the development of prion disease. More than 30 point mutations have been link with inherited prion disease, these are from all parts of the Prion primary sequence, not just near V127. Furthermore, PrPSc core beta-sheet structure is in large parts of the sequence not just near 127.*
- While the reviewer is correct in that the PrP^{Sc} core β-sheet structure appears to extend over much of the structured region of PrP^C, this region of the protein is however extremely important in the development of prion disease, as detailed in the introduction and throughout the manuscript. Heterozygosity at residue 129 provides relative protection against acquired, sporadic and some inherited prion diseases, and also influences the propagation of particular prion strains. The residue 127 mutation has been found solely in unaffected individuals in the kuru-affected population of Papua New Guinea, and provides complete resistance to prion disease in transgenic mice (Asante *et al.*, 2015). G131V and S132I mutations, located in close proximity are also associated with inherited prion disease. This study also highlights that, although a number of human disease-associated mutations may be distant in primary sequence from this region, several are spatially close in PrP^C and appear to interact either directly with residues 127 or 129, or indirectly through residues which themselves have been perturbed by these polymorphisms. These include disease-associated mutations in the PrP β2-α2 loop (D167N), helix 2 (D178N) and helix 3 (V210I, E211Q, Q212P, Q217R, E219K). **We have noted the reviewer's point in the text.**
6. *The reader is left feeling that the studies of the PrPC structure and dynamics do not clearly identify the reasons for the protective nature of the G127V mutation. See for example the vagueness of the conclusion within the abstract. Can this be made firmer.*
- **We have emphasised in the abstract and discussion sections the role that dimerisation may have in prion propagation, and how the V127 might be acting through strengthening the dimer interaction.**

Other minor suggestions:

7. *Substitutions to Valine (a large bulky hydrophobic residue) will typically promote beta-sheet and self-association required for amyloid formation There are numerous examples of this in familial prion disease GSS including P102L, P105L, A117V and G131V all of which mutate to V/L and promote prion disease. I believe this is worth a mention as it further highlights G127V is an interesting exception.*

- **We have added this interesting point to the results section, especially in the light of the fibrilisation data presented.**

8. Line 75, - Ref typo

- **Corrected**

9. Line 79, 2009 not recently

- **We have deleted the offending adverb.**

10. Line 162, posit = suggest

- **We have replaced posit with the reviewer's suggestion.**

11. Line 331, needs a reference

- **We have added references for studies of the thermodynamic stabilities of a number of prion strains.**

12. Line 372 Ref typo

- **Corrected**

We hope that these substantial additional data and edits adequately address the comments of your reviewers and that the revised manuscript is now suitable for publication.

With best wishes

John Collinge

REVIEWERS' COMMENTS:

Reviewer #1 (Remarks to the Author):

In the revised form of the manuscript "Structural effects of the highly protective V127 polymorphism on human prion protein" the authors tried to address all the comments and concerns raised by both the reviewers except point 3 raised by reviewer 1. I can understand that the growing protein crystals are not an easy job and need more effort. It might not be possible to complete these experiments in the review timeline. Though the authors compared their anti-body bound structure with the other published structure in the absence of antibodies that indirectly answers that point as well. The revised manuscript scientifically sounds good and all the results are supported by the experimental data.

Reviewer #2 (Remarks to the Author):

The authors have done a good job responding to my suggestions